# Modifiable and Non-Modifiable Risk Factors for the Development of Non-Hereditary Pancreatic Cancer

**DOI:** 10.3390/medicina58080978

**Published:** 2022-07-22

**Authors:** Marek Olakowski, Łukasz Bułdak

**Affiliations:** 1Department of Gastrointestinal Surgery, Medical University of Silesia, Medyków 14, 40-752 Katowice, Poland; olakom@mp.pl; 2Department of Internal Medicine and Clinical Pharmacology, Medical University of Silesia, Medyków 18, 40-752 Katowice, Poland

**Keywords:** pancreatic cancer, non-hereditary, risk factor, lifestyle, prophylaxis

## Abstract

Pancreatic cancer is becoming an increasing healthcare concern. Though it is a 14th most common cancer worldwide, its incidence is steadily rising. Results of currently available therapies are still not satisfactory. Therefore, great attention should be put on the identification and reduction of risk factors for pancreatic cancer. A thorough up-to-date review of available data on the impact of well-established and novel risk factors of pancreatic cancer development have been performed. Several risk factors associated with lifestyle have significant impact on the risk of pancreatic cancer (i.e., smoking, obesity, alcohol consumption). Physicians should also be aware of the novel findings suggesting increasing role of microbiome, including viral and bacterial infections, in the development of pancreatic cancer. A growing body of evidence suggest also an increased risk during certain occupational exposures. In general, lifestyle seems to be a major contributor in the development of pancreatic cancer. Special attention should be given to individuals with a vicious cluster consisting of metabolic syndrome, tobacco smoking and alcohol consumption. Physicians should urge patients to comply to healthy diet, cessation of smoking and moderation of alcohol consumption, which may halve pancreatic cancer incidence. Further studies are warranted to explore the potential use of therapeutic approach on novel risk factors (e.g., microbiome).

## 1. Introduction

Pancreatic cancer (PC) is the 14th most common human cancer worldwide. In 2020, there were 495,773 new PC cases which was accompanied by 466,003 cancer deaths worldwide, which translates to 2.6% of new cancer cases [1]. PC belongs to a small group of cancers with a steadily increasing incidence rate (1–2% per year). Therapeutic outcomes of PC remain unsatisfactory. Epidemiological data, even from high-income countries, show that only 11.5% of patients survive 5 years after established diagnosis. [2]. As a result of improving outcomes of most common cancer types (e.g., breast cancer) it is estimated that PC will become a third leading cause of deaths in Europe [3].

PC risk factors can be divided into two major groups: non-hereditary risk factors, which depend on lifestyle and environmental influences on the human body, and hereditary risk factors, which are determined by familial or spontaneous and germline or somatic mutations. Among the non-hereditary risk factors for PC, two subgroups can be distinguished: modifiable (e.g., smoking, obesity) and non-modifiable risk factors (e.g., age, gender) (Figure 1) [4]. In the current review we present the non-hereditary causes of PC, while hereditary are covered elsewhere.

## 2. Materials and Methods

Electronic database (PubMed/MEDLINE) search was conducted using selected terms and combinations of term: “pancreatic cancer”, “risk factors”, “modifiable”, “non-modifiable”, “prophylaxis, non-hereditary”. Obtained results were narrowed down to full-text papers published in English. Additionally papers selected for the initial search were limited to time period between the beginning of 2010 and the end of 2021. On the next step selection of manuscripts that were deemed relevant to the topic was performed. Additionally, several manuscript essential to the topic were included despite their publication prior to 2010.

## 3. Modifiable Risk Factors for PC

### 3.1. Tobacco Smoking

Pooled analyses of case-control studies by the International Pancreatic Cancer Case-Control Consortium (PanC4) [5] and Pancreatic Cancer Cohort Consortium (PCCC) [6] groups confirmed that smoking is associated with an approximately 2-fold increased PC risk. This risk increases with the number of cigarettes smoked (≥30 cigarettes per day), smoking duration (≥50 years) and cumulative smoking burden (≥40 pack-years). People who smoke less heavily over a longer period of time have a higher risk than those who smoke more heavily over a shorter period. It is only about 15–20 years after smoking cessation that PC risk decreases to the level found in non-smokers.

A pooled analysis by Koyanagi et al. [7] showed an increased PC risk for current smoking compared with never smoking in both males and females. Significantly increased risk for former smoking and small cumulative dose of ≤20 pack-years were observed only in females, regardless of environmental tobacco smoke exposure. Trend analysis showed a statistically significant 6% increase in PC risk for every 10 pack-years in men (aHR, 1.06; 95% CI, 1.03–1.10) but not in women (aHR, 1.06; 95% CI, 0.92–1.22). Only in males, risk became comparable with never smoking participants after 5 years of smoking cessation. In females, no risk attenuation by smoking cessation was observed.

A meta-analysis by Lugo et al. [8] including 78 epidemiological studies showed that an increased PC risk was noted already in participants smoking a low number of cigarettes and up to 30 cigarettes/day. Similarly, the PC risk increased after a few years of smoking up to 30 years, and it rapidly decreased with increasing time since quitting. 

A meta-analysis by Iodice et al. [9] including 82 epidemiologic studies showed that the overall PC risk was 1.74 (95% CI 1.61–1.87) and 1.2 (95% CI 1.11–1.29) in the current and former smokers, respectively. The PC risk was 1.47 (95% CI 1.17–1.83) and 1.29 (95% CI 0.68–2.45) for current and former pipe and/or cigar smokers, respectively. For former cigarette smokers, the increased risk was noted for a minimum of 10 years after cessation. Concluding, the authors noted a 75% increased PC risk compared to non-smokers, and persistence of this risk for a minimum of 10 years after smoking cessation. 

A meta-analysis by Ben et al. [10] indicated the association between cigarette smoking and an increased total mortality in PC patients. This impact of smoking did not depend on alcohol consumption, body mass index, and history of diabetes but was modified by tumor stage and study settings. Dose-response (including smoking intensity, cumulative amounts of cigarettes smoked, and duration of smoking) associations between smoking and PC mortality were also observed.

A large study [11] on a European population showed that also exposure to passive smoking at the workplace increases the risk of PC of a never smoker. Likewise, early childhood exposure to passive smoking increases the risk of PC by 2.5 times. This study enrolled 465,910 participants. An increased PC risk was noted in current smokers compared to never smoking participants (HR = 1.71, 95% CI = 1.36–2.15), and risk increased with greater intensity and pack-years. Former smokers who quit smoking for less than 5 years had an increased PC risk, but it was comparable to never smoking participants after quitting for 5 years or more. PC risk was increased among never smokers daily exposed to smoking (for many hours) during childhood and exposed to passive smoking at home and/or work. Thus, this study confirmed that both active cigarette smoking, as well as passive smoking, is associated with increased risk of pancreatic cancer [11]. Similarly, maternal cigarette smoking and fetal exposure to tobacco smoke increases the risk of PC by 1.5 times in a child’s adult life [12].

Despite smoking prevalence experiencing a steady decline over the last 10 years among men and women, 9% and 4% respectively, the smoking habit is still widespread. According to the WOBASZ II survey (2013–2014), about 30% of men and 21% of women were identified as regular smokers [13].

Components of tobacco smoke induce a number of pathophysiological changes in the cells of the smoker’s pancreas. They disrupt the physiological process of pancreatic enzyme secretion. Carcinogens present in tobacco smoke affect the microenvironment causing chronic inflammation (via overproduction of pro-inflammatory cytokines), disrupting functions of the immune system (lack of CD8+ T lymphocytes) and stimulating fibrosis (characterized by desmoplastic reactions mediated by pancreatic stellate cells). Components of tobacco smoke interact strongly with DNA, resulting in mutations (i.e., KRAS) and genomic instability, consequently directing the cell towards carcinogenesis [14].

Cigarette smoking is also associated with a statistically significant reduction of the survival of PC patients. This observation applies to patients with high plasma cotinine levels, which is the primary metabolite of nicotine in the blood [15]. In Alkashash’s et al. [16] retrospective analysis of 95 patients with PC, smoking at time of diagnosis, increased risk of death by three folds (HR 3.05, 95% CI, 1.45–6.40).

Furthermore, the type of tobacco product may be related to PC risk. For example, cigar smoking has been shown to increase the risk of PC, but not pipe smoking [17]. E-cigarettes are increasingly replacing traditional cigarettes containing tobacco leaves. They deliver heated nicotine to the human body with fewer of the chemicals present in typical tobacco smoke. Currently, little data are available on the short- and long-term health effects of e-cigarettes [18]. Nevertheless, experimental studies [19] in animals have shown that even pure nicotine induces genetic changes that promote PC development. 

It has been shown that the substances of e-cigarette aerosols may induce alterations of epigenome in human and experimental models, which persist even from fetal into old age. The consequences of these epimutational changes might lead to many diseases, such as chronic obstructive pulmonary disease (COPD) and lung cancer. Therefore, the health risks from e-cigarette smoking should not be ignored [20].

In the first prospective cohort study [21] performed in Iran, it was shown that long-term opium smokers had more than a 3-fold increased risk of developing PC compared with those who had never taken the drug. In 2019, Al-Awwad et al. [22] determined dietary and lifestyle factors, including smoking, for Jordanian patients with PC. In this study, a number of cigarettes smoked per day was 10.8 ± 1.6 for patients with PC, and 9.0 ± 0.9 for controls. Moreover, the total number of smoking participants (including current, previous, and passive smokers) was significantly higher for PC (46.5%) than controls (39.5%). Thus, this study confirmed that smoking can increase PC risk. 

A case-control study by Silverman et al. [23] showed a significant, 70% increase risk of PC in smokers compared with nonsmokers. They also observed a negative trend in PC risk with increasing years of quitting smoking. In smokers quitting for more than 10 years, about a 30% reduction in risk relative to current smokers was noted, whereas there was no risk reduction in quitters of 10 years or less. Switching from non-filtered to filtered cigarettes did not appear to decrease risk. Compared with non-smokers, patients smoking only filtered cigarettes had a 50% increased risk and smoking non-filtered cigarettes—a 40% increased risk.

However, there are also studies showing contradictory data to those described above. A clinic-based control study by Lea et al. [24] including a population of the San Francisco Bay Area recruited patients between 2006 and 2011, and revealed no significant association between cigarette smoking and PC. No association was found for either former (OR = 0.85, 95% confidence interval [CI] = 0.66–1.1) or current cigarette smoking (men: OR = 1.0, 95% CI = 0.60–1.7; women: OR = 1.2, 95% CI = 0.73–2.1). No dose-response relationships were noted with number of cigarettes per day, smoking intensity, duration, or years since last smoked. Compared to the study performed earlier (1995–1999) in the same area the number of smokers was significantly decreased. Interestingly, researchers from the same site in their earlier study performed also in the area of San Francisco Bay showed that smoking is associated with increased of PC [25]. Importantly, the increased risk of PC was not observed in smokers who had quit for ≥10 years.

In conclusion, the majority of the prospective studies and meta-analyses shows positive association between smoking and PC risk. The impact of current, former as well as active and passive smoking increase the PC risk. Some differences were noted with regard to the type of used tobacco products (i.e., cigarettes, pipes, cigars), which may reflect the actual amount of tobacco smoke that is absorbed to organism. 

### 3.2. Ethanol

Evidence supporting a causal relationship between alcohol consumption and PC has so far been equivocal [26,27,28,29].

In 2018, the European Prospective Investigation into Cancer and Nutrition (EPIC) published the results of a multicenter prospective study showing a statistically significant increase in PC risk in men who consumed large volumes of alcohol (>60 g of ethanol per day) during their lifetime as compared to men who were not heavy-drinkers (<60 g/day). Conversely, no correlation was observed between alcohol consumption and PC in women. In a previous analysis of the association between the type of alcohol consumed and cancer risk, there was a stronger risk for beer and liquor than for wine consumption [30].

Interpretation of the results of studies assessing the effect of alcohol consumption on the PC development is complicated by frequent combination of heavy alcohol consumption with smoking habits [31]. A case-control study by Rosato et al. [32] indicated that 13.6% of PC were attributable to smoking, and 13.0% were associated with heavy alcohol drinking. Concomitant alcohol consumption and smoking elevated the PC risk up to 25.7%. Another Italian multicenter study [33] showed that alcohol consumption was associated to increased PC risk. The authors noted that PC risk was 4.3-fold higher in heavy smokers (≥20 cigarettes/day) and heavy drinkers (≥21 units/week) compared to never smokers drinking < 7 drinks/week, which was compatible with an additive effect of these exposures.

A comparative American study [34] assessed alcohol as a risk factor of PC according to the race (white vs. black). Compared to whites, blacks presented significantly higher ORs associated with heavy alcohol drinking (≥57 units/week) in males (*p* = 0.04) and with moderate-to-heavy drinking (≥8 drinks/week) in females (*p* = 0.03).

In summary, there are only limited number of studies regarding the association between alcohol consumption and PC. So far, majority indicates alcohol as a risk factor for PC, especially in heavy-drinkers. There are minor differences in the impact of alcohol consumption based on the type of alcoholic beverage. But there several confounders in those analyses, due to various dietary and social habits. Alcohol-related PC risk increases with the simultaneous smoking. The subject warrants furthers exploration. 

### 3.3. Dietary Habits

It is well known that food can predispose to the development of certain diseases, but proving connection between individual dietary component and the risk of cancer development is difficult [35].

#### 3.3.1. Red Meat

A meta-analysis of 11 prospective studies [36] confirmed a positive correlation between pancreatic cancer incidence and consumption of large amounts of red (>120 g/day) or processed meat (>50 g/day). Furthermore, men who consume approximately one portion (50 g) of processed meat per day during their lifetime have been shown to have a 19% increased PC risk. Curiously, red meat consumption was associated with an increased PC risk in men, but not in women. 

In 2017, Zhao et al. published a large meta-analysis [37] of data from 28 studies (case-control studies and cohort studies). The study showed that red and processed meat consumption was positively associated with PC risk in case-control studies, whereas no such association was observed in the data from the cohort studies. 

A prospective cohort study [38] from Boston University found that a consumption of red and unprocessed meat and saturated fats was associated with an increased PC risk in African-American women over the age of 50. 

Meat preparations which are preserved with nitrites (potassium nitrate) may contain nitrosamines (NDMA), which are potent carcinogens. These chemical compounds are mainly formed by reactions between secondary and tertiary amines and nitrites. Such reactions can occur during thermal processing of food (at temperatures > 130 °C) and directly in the human body when food containing nitrates/nitrites comes into contact with hydrochloric acid in gastric juice. In a large controlled clinical study [39], a probable association between the consumption of foods containing two nitrosamine derivatives and PC development was confirmed. 

#### 3.3.2. Carbohydrates

Unlike in Europe, over the past 10–20 years, fructose has largely replaced sucrose as a sweetener in soft drinks in the United States. The third National Health and Nutrition Examination Survey [40] found that more than 10% of Americans’ daily calories come from fructose. A comprehensive meta-analysis of prospective studies confirmed that fructose was the only sugar that increased the PC risk. It is estimated that for every 25 g of fructose consumed per day, the lifetime risk of PC increases by 22%. 

#### 3.3.3. Lipids

Epidemiological studies on the relationship between high-fat diets and PC lack convincing evidence, however, most of them suggest that PC incidence increases in countries with a high-fat diet, especially rich in saturated fatty acids [41].

A cohort study [42] based on data obtained from the NIH (National Institutes of Health) in the United States found that the risk of PC increases if high fat intake occurs during adolescence and midlife (45–60 years). 

#### 3.3.4. Diet Decreasing the Risk of PC

A large Chinese study [43] showed that tea intake halved the PC risk. Reduced vegetable consumption led to a higher PC risk, but a significant protective effect was found for for regular fruit consuption (1–2 times/week vs. more than 3 times/week; OR = 1.73, 95% CI: 1.05–2.86). Contrary to the protective effects of vegetable dishes, high meat intake was associated to a higher PC risk (OR = 0.59 for consumption of 1–2 times/week vs. more than 3 times/week; 95% CI: 0.35–0.97).

It is important to note that most articles that have appeared in recent years on the relationship between specific dietary components and PC risk are often contradictory. Therefore, it has been proposed that specific dietary patterns, rather than individual nutrients, should be analyzed in such studies [44].

In conclusion, red meat, simple carbohydrates (predominantly fructose), and high-fat diet increase the PC risk. A protective properties against PC were reported for tea, vegetables and fruits.

### 3.4. Obesity

Obesity is strictly connected with dietary habits and results in the buildup of fat in adipose tissue due to caloric excess. Abdominal accumulation of lipids is especially important as a risk factor for PC. According to an American Cancer Society (ACR) study, the PC risk in obese individuals in both sexes was twice as high as in individuals with a normal BMI [45].

A cohort study [46] performed by EPIC on a European population confirmed that an increase in body weight resulting in increased BMI correlated with higher risk of various cancers, including PC in men. A meta-analysis [47] of prospective cohort studies found that a 5 kg/m^2^ increase in BMI was associated with a 12% increase in PC risk. 

The molecular basis of the process by which dysfunctional subcutaneous and visceral adipose tissue promotes PC growth is complex. Excessive fat accumulation causes hypoxia and mild inflammation in adipose tissue, resulting in the release of proinflammatory cytokines and adipokines. Elevated levels of circulating leptin can lead to PC invasion through activation of the JAK2/STAT3 signaling pathway in cancer cells. Other adipokines, including resistin, lipocalin-2, apelin and visfatin, may also promote the growth and progression of pancreatic cancer cells. In addition, inflammation of visceral adipose tissue induces insulin resistance, which promotes systemic secretion of insulin and IGF-1. Activation of insulin receptor (IR) and insulin-like growth factor-I receptor (IGF-IR) increases pancreatic cancer cell proliferation through activation of PI3K/mTOR and MAPK/ERK signaling pathway [48].

Shyam et al. [49] studied the association of both conventional (BMI, waist and hip circumference and waist–hip ratio) and novel (UK clothing sizes) obesity indices with PC risk in the UK women’s cohort study (*n* = 35,792). During the 654,566 person-years follow up, there were 136 incident PC cases. Baseline BMI was a significant risk factor for PC. 

According to an interesting, randomized controlled trial [50], a low-fat diet was associated with reduced PC incidence in overweight or obese women. But such an association was absent in women with a normal BMI. This study suggests the possibility of introduction of low-fat diet in overweight and obese patients as a preventive measure against PC.

Concluding, obesity is associated with an increased PC risk and it seems that abdominal accumulation is the culprit. On the bright side, dietary interventions may blunt the risk of PC in overweight and obese subjects.

### 3.5. Type 2 Diabetes Mellitus

Type 2 diabetes mellitus (T2DM) is the third most important modifiable risk factor for PC after smoking and obesity. In many cases it results from and is aggravated by bad dietary habits. Epidemiological studies show that long-standing type 2 diabetes mellitus (LST2DM) is associated with a 1.5- to 2.0-fold increase in PC risk [51]. Approximately, 85% of PC patients have diabetes or hyperglycemia. A population-based cohort [52] of PC patients (*n* = 219), revealed that 42% of them had diabetes and 52% had new onset diabetes (NOD), 13% had advanced pre-diabetes, 21% had an abnormal FGL (fasting glucose level), and only 9% had a normal FGL. 

Frequently, however, diabetes abruptly manifests itself 2–3 years before the PC diagnosis. Patients with newly diagnosed diabetes have as much as a 5–8 fold increased PC risk within 1–3 years of onset [53]. The ENDPAC (Enriching New-Onset Diabetes for Pancreatic Cancer) model was developed to determine PC risk in patients with NOD. Based on glycemic status, the ENDPAC model can identify 75% of pancreatic cancer patients with NOD 6 months before the cancer diagnosis [54]. 

A Chinese study found that not only diabetes but also hyperglycemia were associated with an increased PC risk. Among those without diabetes, for every 18 mg/dL blood glucose level elevation, there was a 15% increase in PC risk [55]. Hyperglycemia leads to increased synthesis of highly reactive carbonyl compounds (‘carbonyl stress’) and causes a damage of various molecules in cells, forming end products known as AGEs (advanced glycation end-products) that can accelerate PC growth. These results suggest that carbonyl stress is involved in cancer development and growth and potentially links diabetes and PC [56].

Insulin released by β-cells arrives with the blood via the intra-pancreatic portal circulation to the follicular and ductal cells adjacent to the islets. PC development on the basis of diabetes depends on the long-term persistence of high intra-pancreatic insulin levels that can promote survival and proliferation of any transformed cells that may arise in pancreas. Thus, hyperinsulinemia, particularly intra-pancreatic hyperinsulinemia, may likely contribute to the observed increased PC risk [57] and the mechanism responsible for that may to some extent rely on the above-mentioned unspecific stimulation of IGF-1 receptors by insulin at high concentration. Additionally, insulin treatment was associated with a significantly higher PC risk than therapy with other hypoglycaemic drugs [58]. However, the International Pancreatic Cancer Case-Control Consortium [59] assessed the effect of insulin on PC risk in patients with T2DM as unclear; a short period of insulin use (<5 years) significantly increased PC risk, while a longer period (>15 years) did not. A recent analysis [60] showed that PC risk associated with insulin therapy is dependent not on the type of hypoglycaemic drug as on the diabetes subtype. PC risk is higher in patients with the NOD diabetes subtype than in the LST2DM subgroup.

An increased risk of PC has been observed in patients using sulphonylurea derivatives or insulin [61]. However, one should consider that most of the available studies evaluating the effect of antidiabetic drugs on cancer development have significant limitations because they do not include confounding factors that may affect results. Therefore, despite the large number of studies suggesting an association between cancer and some antidiabetic drugs, the consensus states that the data are not significant enough to change therapeutic approach in diabetes, especially with regard to insulin [62].

In conclusion, currently it is clear that T2DM is associated with an increased PC risk, but further studies regarding this subject are required. Those studies should focus on the impact of antidiabetic drugs on PC development. It seems necessary, due to the advent of novel antidiabetic drugs, especially GLP-1 analogs.

### 3.6. Metabolic Syndrome 

Metabolic syndrome (MS) is a clinical entity that is connected with an increased cardiovascular risk. There are several definitions of MS, but in general it comprises of visceral obesity, hyperglycemia, features of atherogenic dyslipidemia, and hypertension [63].

Epidemiological studies have shown that MS and its components may independently or combined increase PC risk [64]. A cohort study [65] using the UK Biobank database showed that in addition to MS, two of its components were independently associated with PC risk: greater abdominal waist circumference and blood glucose levels. A study [66] using the Korea National Health Information Database (NHID) found that a significant increase in PC risk (HR; 1.64) occurs only when a single person has ≥4 MS components. A prospective large study by Johansen et al. [67] suggested a possible association between abnormal glucose metabolism and PC risk.

Recently, nationwide cohort study, including 8,203,492 patients, by Park et al. [68] demonstrated an association between MS and an increased PC risk in the general population. In addition, the authors noted that this relationship was different depending on changes and persistence of MS status. The highest risk of PC was noted in patients with persistent MS. The risk of PC was higher in patients with the MS-recovered patients compared to the MS-free patients, but lower compared to the MS-persistent patients. In conclusion, recovering from MS can reduce the risk of pancreatic cancer and changes in the MS status leads to differences in the PC risk.

Several studies have shown association between metabolic syndrome and PC. Taking into account an increased incidence of both MS and PC, a targeted actions to reduce the number of metabolic components may have a beneficial effect on decreasing the risk of pancreatic cancer.

### 3.7. Acute Pancreatitis (AP) 

A newly discovered risk factor for PC is a history of AP that occurs 1–2 years before PC diagnosis. A cohort study [69,70] of 2 Scandinavian populations showed that individuals with a history of AP had doubled the risk of developing PC than those who did not undergo such a disease. 

The risk of PC is substantially increased during the first few years following an AP diagnosis and declines gradually over time, reaching a level comparable to that of the pancreatitis-free population after >10 years of follow-up. These findings may suggest a delay in the diagnosis of pre-existing pancreatic cancer, which was initially presented as acute pancreatitis [69].

A laboratory assay that may indicate the presence of PC in a patient with biliary pancreatitis is the concentration of the marker Ca19-9. It should be noted that the level of Ca19-9 in the blood of a person with cancer is usually significantly elevated, contrary to a slight increase in Ca19-9 during biliary pancreatitis [71].

Earlier epidemiological studies were limited to analyses on either diabetes or pancreatitis as a single cancer risk factor. Only recently, a study from New Zealand [72] has provided evidence that both diseases may synergistically increase PC risk. In a study of 139,843 people, of which 913 were diagnosed with PC, the percentage of patients diagnosed with cancer was as follows: 3.1% of patients with diabetes after AP, 2.3% with type 2 diabetes before AP, 2.0% of patients diagnosed only after AP, and 0.6% of patients with T2DM alone without an AP diagnosis. Most notably, only in patients with diabetes occurring before or after AP pancreatic cancer risk was significantly higher than in patients with T2DM alone. In a study [73] of the Dutch Pancreatitis Study Group, it was shown that although the first episode of acute pancreatitis may be causally related to the presence of PC, an increased PC risk is mainly observed in patients who progress from acute to chronic pancreatitis. Further studies are needed to confirm the relationship between AP and the risk of PC.

### 3.8. Chronic Pancreatitis (CP)

In recent years, there has been increasing evidence of an association between long-standing CP and an increased PC risk. Both pathologies exhibit similar morphological and genetic changes as observed in histopathological samples. The K-RAS oncogene mutation may potentially act as the molecular link between the diseases, as it is present in approximately 90% of PC cells and found in the hyperplastic transformation of pancreatic duct epithelial cells of patients with CP [74]. However, the pathogenesis of PC development based on CP remains unknown. Other factors connecting CP and PC include the macrophages, the maintenance of genome stability, cytokines, nuclear factor kappa B, COX-2 and reactive oxygen species [75].

The risk of PC development based on CP is highest in rare types of pancreatitis such as hereditary or tropical pancreatitis [76]. Malignant transformation in CP occurs more often in elderly patients and long-term heavy smokers (>60 pack-years) [77]. It is estimated that the risk of PC increases with the duration of CP and reaches 4% 20 years after diagnosis [78]. Recent observations [79] show that while the total number of cancer cases based on CP increases with the length of the observation period, the relative risk of cancer in individual patients tends to decrease with the duration of the disease and is highest early after the diagnosis of CP. During the first 2 years PC risk in patients with CP was increased 16-fold, after 5 years by 8-fold and after 9 years by only 3-fold in comparison with the group of patients without CP. Therefore, it is now recommended that patients with newly diagnosed CP should be closely monitored in the first years after diagnosis. 

Based on a retrospective analysis of clinical data [80], 2 subgroups of CP patients at high risk for pancreatic cancer were identified. The first subgroup included patients with low BMI, exocrine insufficiency and CP without a previous episode of AP, and the second subgroup included patients with elevated BMI and T2DM. Early identification of patients from both subgroups provides an opportunity to direct them to increased surveillance for early detection of pancreatic cancer. 

Concluding, CP is a risk factor for PC. Patients with newly diagnosed CP should be closely monitored especially in the first years after diagnosis. The PC risk is highest in rare types of pancreatitis (such as hereditary and tropical pancreatitis), but the occurrence of malignant transformation in CP significantly increases also in the elderly population and long-term heavy smokers.

### 3.9. Microbiome

The intestinal microbiome is an essential element of the organism crucial for maintaining host homeostasis. Microbial dysbiosis and impaired epithelial barrier function, which leads to bacterial translocation, are both inducers of several diseases including neoplastic transformation [81]. As a parenchymal organ, the pancreas has no microbiome. It appears, however, that in pathological conditions local microbiota, appears in the gland [82]. This was verified in a study [83] using the technique of measuring bacterial DNA in tissues (16S rRNA gene sequencing), which revealed an increased presence of certain gastrointestinal bacteria (*Proteobacteria*—45%, *Bacteroidetes*—31%, and *Firmicutes*—22%) in the tumour-lesioned pancreas of both mice and humans. The bacterial translocation to the pancreatic gland affects the local immune system of the pancreas, possibly contributing to a faster progression of cancer. In a study [84] of the microbiome from pancreatic and duodenal tissue samples collected during surgery in humans, many bacterial species were detected in both cancer and non-cancer patients. Analysis of their composition revealed the presence of bacteria inhabiting the oral cavity and periodontal disease. The methods of samples taking is very important and precautions should be taken to avoid contamination by extra-tumoral and environmental bacteria and several controls should be used. The channels that microbes use to reach the pancreas have been not clearly indicated. The pancreas is connected to the digestive tract via duodenum. Therefore, oral microbes can enter the pancreas through the digestive tract. On the other hand, it is well known that oral microorganisms easily enter the blood, leading to bacteremia. Therefore, oral microorganisms may also enter the pancreas through blood circulation [85].

The microbiome may be involved in a number of pathophysiological pathways. Firstly, the development of the local inflammatory state in the pancreas initiates the release of many pro-inflammatory factors, i.e., growth factors, lipopolysaccharides, TGF-β, TNF-α, which activate signaling pathways in cells (TLR-4—Toll like receptor 4, mTOR, NF kappa B/MAPK) potentially promoting the development of PC. Secondly, there is a reduction in the host immune response as the interaction between macrophages and T cells is inhibited and Th cells activation is impaired. Thirdly, there are changes in the metabolism of carbohydrates, amino acids, lipids, synthesis of vitamins and other nutrients that lead to diabetes and obesity, which increase PC risk [85]. Ablation of the microbiome or certain targeted changes in its composition is considered as a target against early and invasive course of pancreatic ductal adenocarcinoma [83]. 

Microorganisms isolated from the microbiome can serve as diagnostic markers for PC and as valuable tools to assess chemotherapy and immunotherapy efficacy [85]. Therefore, the knowledge on the microbiome in PC is important for the PC detection using non-invasive fecal sample collection as well as targeted oncological therapy. Further investigations are needed to assess pancreatic microbiome and the routes of bacterial translocation into the pancreatic gland.

Data regarding the microbiome participation in the development of PC are very interesting, but scarce. Therefore further studies are required to analyze association between bacterial translocation and PC risk. Currently, this relationship is not clearly determined, but its potential introduction in the early diagnosis and targeted treatment seems plausible.

### 3.10. Periodontal Diseases

Numerous clinical studies performed over the past decade have shown an association between oral cavity diseases and the development of various human cancers, including pancreatic cancer [86]. Specific bacterial infection can cause the development of periodontal diseases. Among various microorganisms that colonize the oral cavity, 3 strains (*Porphyromonas*, *Tannerella* and *Actinobacillus)* have been identified as key in the development of periodontitis [87]. A potential link between the oral microbiome and the probability of developing PC was explored and it was observed that people with high levels of antibodies to the periodontal tissue-destroying bacterium *Porphyromonas gingivalis* ATTC 53978 were at more than twice the risk of PC [88]. In addition, a reduced risk of PC was observed in subjects with elevated levels of antibodies against certain commensal (non-pathogenic) oral bacteria. In a prospective cohort study [89], it was observed that while the presence of oral pathogens like *Porphyromonas gingivalis* and *Aggregatibacter actinomycetemcomitans* was associated with a higher risk of PC, the presence of bacteria like *Fusobacteria* and *Leptotrichia* reduced this risk. A recently published meta-analysis [90] involving 8 clinical trials showed that both periodontal disease and lack of teeth are risk factors for PC. The relative risk of PC for those with periodontitis was found to be 1.74 times higher and for those without teeth, 1.54 times higher compared with people with teeth and without oral cavity disease.

In conclusion, periodontitis and periodontal disease seem to be associated with an increased PC risk, which probably strongly relies on microbiome itself and induced inflammatory state. However, this theory should be confirmed by further studies.

### 3.11. H. pylori Infection and Peptic Ulcer Disease

Infection of the gastric mucosa with *Helicobacter pylori*, especially the cytotoxic strain (CagA+), is a known risk factor for peptic ulcer disease and gastric cancer. According to some researchers it may also contribute to the development of PC [91]. A study of the microbiota present in the duodenum of patients with pancreatic head cancer found a higher frequency of infection and mucositis around the papilla of Vater caused by *H. pylori* [92]. The results of single studies on the increased PC risk in individuals with specific antibodies to *H. pylori* are conflicting. Some confirmed such a connection [93] whereas others opposed it [94]. However, based on meta-analyses a statistically significant association between *H. pylori* infection and PC development was suggested [95,96].

A hypothetical reason for the carcinogenic effect of *H. pylori* on the pancreas is that it multiplies the potency of the toxic effect of N-nitrosamines, which enter the body with tobacco smoke or food. This effect is modulated by the characteristics of the bacterium resulting from its virulence and the interaction between the microorganism and the host immune system [97]. Interestingly, proton pump inhibitors (PPIs) used in the eradication of *H. pylori*, can cause hypergastrinemia, which may promote the PC development and progression. Thus, there are indications to be cautious when using this group of drugs in patients at high risk of PC [98].

Patients that underwent gastrectomy for peptic ulcer disease using the Billroth II resection more than 20 years earlier were found to have had a significantly increased PC risk [99].

Concluding, a correlation between *H. pylori* infection and PC risk is not clearly determined and it requires further studies, but its eradication might become another indication to prevent carcinogenesis in the future.

### 3.12. HBV and HCV Infection

Recent meta-analyses [100,101] suggest a higher risk of PC in HBV-infected (1.39 times) and HCV-infected (1.5 times) individuals. PC risk is somewhat modified after taking into account additional epidemiological data such as diabetes, chronic pancreatitis, alcohol consumption and smoking in the case of HCV-infected individuals, or the region in which the study population resided (Europe vs. Oceania) in the case of HBV-infected individuals. Precise mechanisms responsible for increased risk of PC in viral hepatitis are not fully explained. There are studies showing the presence of HBV core antigen in acinar cells [102] confirming viral invasion of pancreatic cells, which might similarly to hepatocytes be responsible for chromosomal instability [103]. Another factor that may contribute to elevated PC risk is connected with more common occurrence of chronic pancreatitis in patients with hepatotropic viral infection [104]. Nevertheless, further studies are necessary to substantiate potential causal link between those viral infections and PC.

### 3.13. Cholecystectomy

Recent epidemiological studies support a potential weak association between the occurrence of gallstones [105,106], cholecystectomy [106,107] and PC risk. The reason for that is not clear, it might stem from slightly increased biliary tract inflammation risk.

### 3.14. Occupational Exposure

The results of studies on the effect of occupational exposure on PC are uncertain. Many studies link the occurrence of PC to occupational exposure of various chemicals, i.e., pesticides [108], chlorinated and aromatic hydrocarbons [109], and metals (i.e., lead, cadmium and arsenic) [110].

These substances may reach the pancreas through the bloodstream or bile reflux and exert genotoxic effects on its cells, but the exact molecular mechanism that can lead to the development of PC is not yet understood [109].

Another factor that is considered is the disruption of the daily rhythm that accompanies shift work and may play a role in carcinogenesis. Recent studies [111] suggest that the light at night (LAN) that accompanies workers during the night work disrupts the daily rhythm and may predispose them to PC. Contrary to people who work one shift, those who work at night are more likely to be obese and have T2DM, which are two major risk factors for PC.

## 4. Non-Modifiable Risk Factors for Pancreatic Cancer

### 4.1. Gender

According to the GLOBOCAN, pancreatic cancer is more common in men (5.7 per 100,000) than in women (4.1 per 100,000) [1]. Findings suggest that differences in environmental exposures and genetic variations are possibly the reason for the increased incidence of PC in the male population [112]. Hypotheses that female sex hormones may protect women from developing PC are unconfirmed [113]. Gender differences in PC incidence seem to gradually diminish with age progression. In a recently published analysis [114], no statistically significant differences in incidence, prevalence and mortality were observed between the two sexes; however, the value of disability- adjusted life years (DALY) was higher in men than in women with borderline statistical significance (*p* = 0.047).

### 4.2. Age

Pancreatic cancer, like most other cancers, frequently occurs in older people. The peak of its incidence falls on the period between the 6th and 8th decade of life. Only in about 10% of cases PC is detected below the age of 50. In older patients, the incidence rate of PC increases rapidly to 9.8 per 100,000 annually at the age 50–54 years and to 57 per 100,000 annually in 20 years older population [115].

Data from 13,131 PC patients (30–95 years) showed that the average age at which cancer was diagnosed in the US was 65–67 years. The disease occurred earlier in men (65.2 years) than in women (66.8). African Americans also developed pancreatic cancer at younger age than Caucasians (62–63 vs. 66 years) [116].

### 4.3. Race

Race is a significant risk factor for PC. Differences in incidence rates between various races are particularly well demonstrated in the US population. In all states except Hawaii, PC incidence and mortality rates are higher in black African Americans than in whites of non-Hispanic origin and other ethnic groups. This observation has been documented for both younger (<50 years) and older adults (>50 years). In addition, almost all racial/ethnic groups showed a higher increase in PC incidence over the 27-year follow-up period. Although the Native American and Alaska Native populations had the lowest incidence of this cancer among all racial/ethnic groups, they showed the greatest rate of increase in PC cases over time. Only in the African American group the rate of increase in pancreatic cancer incidence was stable [117]. In African Americans, PC is more often diagnosed at an advanced stage, and therefore, is frequently assessed as unresectable during surgery [118].

Behavioral factors have a significant influence on the increased incidence of PC among African Americans. Risk factors like smoking, alcohol consumption, obesity, increased consumption of high-calorie foods, T2DM, and low socioeconomic status are more prevalent among African Americans [119].

Differences in PC incidence by ethnic groups may also be due to some molecular differences between these populations. The higher frequency of CDKN2A mutations found in cancer cells in this population compared to other white ethnic groups, or KRAS mutations, may be responsible for the higher PC risk in African Americans [120,121].

In contrast, PC in the Asian-American population stage of the disease at diagnosis is less advanced and usually results in a better prognosis than in the non-Asian population. This means that Asian individuals have a longer overall survival than their non-Asian counterparts. Plausible explanations for survival rate differences between ethnic groups include lifestyle and genetic disparities [122]. A study comparing genetic mutations in pancreatic cancer patients from China, Japan and Western countries found that people of Asian ancestry have different KRAS and p53 gene expression. These results suggest that different races have some genetic and molecular distinctions with respect to each other, which may influence both the incidence rate and outcome in PC [123].

### 4.4. Blood Type

Results of a large cohort study [124] suggest an increased risk of PC in people with blood types A, B and AB. Genetic discoveries that identified the ABO locus on chromosome 9q34 associated with the PC risk [125] suggest that people with blood type O may have a lower PC risk. Other blood components involved in the pathogenesis of PC are isoagglutinins (anti-A and anti-B) that can agglutinate with PC antigens, forming inactive complexes [126]. A large study [127] that included 1.6 million blood donors found an increased PC risk in people with blood type A. Meanwhile, the same study found that blood type B was not associated with an increased risk of PC. Studies on ABO blood groups in PC have not distinguished between A1 and A2 subgroups of blood group A. Data from a study performed by researchers at PanScan [128] and the University Hospital of Bergen (Norway) [129] showed that among all of the common ABO variants, the A1 allele was associated with the highest risk of PC.

A summary of association between risk factors and pancreatic cancer was summarized in Table 1.

### 4.5. Non-Hereditary Genetic Alterations

PC is the result of accumulation of somatic gene mutations such as gain-of-function mutations in proto-oncogenes (i.e., *KRAS*) and loss of function mutation of tumour suppressor genes (i.e., *CDKN2A*/p16, *TP53*/p53, *SMAD4*, *BRCA2* and others). In PC, alterations in promoters, microRNA (miRNA) expression and chromatin structure are also reported. These alterations include methylation regulated gain of function in oncogenes, and loss of function in tumor suppressor gene and DNA repair genes. Hypermethylation of numerous genes was reported in PC. Recurrent somatic mutations encoding epigenome regulators of PC have been identified, including the SWI/SNF complex and the histone-lysine *N*-methyltransferase 2 (KMT2) family. These DNA methylation alterations may be found in the pancreatic juice and therefore might be used for early detection of PC [130].

## 5. Pancreatic Cancer Prevention

Approximately two-thirds of the major risk factors for PC are potentially modifiable, providing a unique opportunity to implement preventive measures [131]. It is estimated that just by reducing smoking in a country’s population, PC could be prevented in about one-third of patients in the future [132]. Similarly, in order to reduce the risk of gastrointestinal cancers, including PC, it is necessary to reduce alcohol consumption [133]. For obese individuals, weight loss, especially permanent and significant as after bariatric surgery, reduces the risk of many cancers and may become a promising intervention for preventing PC [134]. Dietary modifications are crucial in cancer prevention. A cohort study [135] performed in the UK demonstrated that PC mortality was significantly lower in low meat eaters (by 30–45%) and in vegetarians and vegans (by around 50%) compared with regular meat eaters. Furthermore, the Women′s Health Initiative Dietary Modification (WHI-DM) trial [50] revealed that a low-fat diet caused a reduction in PC incidence in overweight and obese women. Similarly, a diet consisting of fruits and vegetables rich in antioxidants has a protective effect against the development of PC, reducing its risk by more than 30% [136]. A 65% reduction in the relative risk of death from PC has been shown in people consuming several servings of dried fruit per week [137]. Additionally, eating large amounts of whole grain products may be beneficial in terms of reducing the risk of PC [138].

Several studies have observed a reduced risk of pancreatic cancer by people taking high doses of vitamin A [139], lycopene, β-carotene and β-cryptoxanthin [140], B6 [141], B12 [142], vitamin C [143], vitamin E [144], vitamin K [145], vitamin D [143] and some micronutrients, e.g., selenium [146]. Supplementation with omega-3 fatty acids present mainly in fish oil and nuts have a protective effect against the development of PC [147].

Pharmacological interventions have also been evaluated. Some studies [148,149] provide evidence suggesting metformin’s anti-cancer effect in patients with type 2 diabetes. Other indicate that aspirin may be used as chemoprevention for people at high risk of cancer, including PC [150,151]. Hormone replacement therapy (HRT) can probably reduce the risk of developing PC as well. In a cohort study [152] on a population of Swedish women, it was observed that the PC risk was reduced by 35% in women using HRT for 1–2 years and by 60% in women using HRT ≥ 3 years compared with women using HRT < 1 year. The type of HRT did not affect the results of the study. It should be kept in mind that neither HRT nor other therapies are recommended as direct prevention of PC. People with atopic diseases (asthma, hay fever) have a reduced risk of developing PC, indicating the involvement of immunological, inflammatory or therapeutic factors that may stimulate and inhibit carcinogenesis [153].

The World Cancer Research Fund/American Institute for Cancer Research (WCRF/AICR) in 2018 proposed an updated set of recommendations [154] for cancer prevention. Specifically, “being physically active” (≥150 min/week), reduced the risk of pancreatic cancer incidence and mortality in the US population. Dietary modifications alone did not produce such beneficial effects as that of physical activity. Other recommendations are as follows: (1) maintaining a normal body weight (BMI 18.5–24.9), (2) eating a food rich in whole grains, vegetables, fruits and beans (vegetables and fruits ≥ 400 g/day), (3) limiting the intake of processed foods high in fats, starches and sugars, especially fast food, (4) limiting the intake of red meat (red meat < 500 g/week), (5) limiting the intake of sweet juices and drinks, (6) alcohol abstinence, and (7) if possible, breastfeeding of newborns.

## 6. Summary

Modifiable and non-modifiable risk factors for PC have been gathered in Figure 1. Table 1 includes data regarding the magnitude of influence of each risk factor on the incidence of PC. Non-hereditary risk factors are divided into two subgroups: modifiable (smoking, drinking alcohol, diet, obesity, type 2 diabetes, metabolic syndrome, pancreatitis, bacterial translocation, infectious and occupational diseases) and non-modifiable (age, gender, race, blood group). Approximately two-thirds of these major risk factors are potentially modifiable, creating a unique opportunity to eliminate them in order to prevent the occurrence of one of the deadliest human cancers. Therefore, this knowledge is important in PC prevention. The association between some risk factors (such as smoking and alcohol consumption) are well known, but the association between the new ones (such as e-cigarettes or bacterial translocation) needs further studies on the large populations.

## Figures and Tables

**Figure 1 medicina-58-00978-f001:**
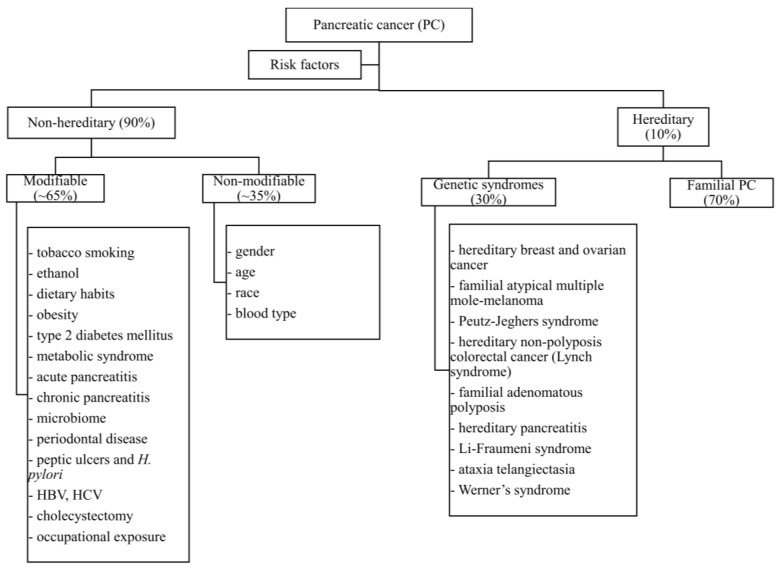
Factors involved in the development of pancreatic cancer.

**Table 1 medicina-58-00978-t001:** Summary of the PC risk factors including the magnitude of risk (provided in brackets).

Category	Modifiable Risk Factor of PC (Risk Level)	Non-Modifiable Risk Factor of PC (Risk Level)
Increased risk	Chronic pancreatitis (3.0–16.0×)	Age (50–70 years old) (6.8×)
	Tobacco smoking (up to 2.5×)	Male sex (1.0–1.4×)
	High ethanol intake (up to 2.0×)	Ethnicity (Afro-Americans vs. Caucasian) (1.3×)
	Acute pancreatitis (2.0×)	Non “0” blood type (1.3–1.7×)
	Type 2 diabetes mellitus (1.5–2.0×)	
	Periodontal disease (1.5–1.7×)	
	Metabolic syndrome (up to 1.6×)	
	HBV/HCV infection (1.4–1.5×)	
	*H. pylori* infection (1.4–1.5×)	
	Frequent red meat consumption (1.2×)	
	Fructose (1.2×)	
	Obesity—per 5 kg/m^2^ (1.1×)	
	Microbiome (pending)	
	Occupational hazard (varied)	
Reduced risk	Tea consumption (0.5×)	
	Diet rich in fruits (0.6×)	
	Allergy—nasal (0.6×)	
	Physical activity (pending)	

## Data Availability

Not applicable.

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
