# Peer review of "Modifiable and Non-Modifiable Risk Factors for the Development of Non-Hereditary Pancreatic Cancer"

_medicina, 2022, doi:10.3390/medicina58080978_

Round 1
Reviewer 1 Report
Well-written paper on an interesting topic. However, several factors are missing, such as:
- - inherited genetic changes: mutations of BRCA1, BRCA2, PALB2, CDKN2A, ATM, MLH1, MSH2, MSH6, PMS2, STK11, and EPCAM. Also KRAS2, p16/CDKN2A, TP53, and SMAD4/DPC4. Genes associated with an increased risk of pancreatic cancer include BRCA1, BRCA2, P16, PRSS1, STK11/LKB1, hMLH1, hMSH2, FANC-C, and FANC-G.
- - Rare inherited conditions: Hereditary pancreatitis, Peutz-Jeghers syndrome, familial malignant melanoma and pancreatic cancer, hereditary breast and ovarian cancer syndrome, Lynch syndrome, Li-Fraumeni syndrome, familial adenomatous polyposis
- - Cirrhosis
The authors should underline the importance of genetic testing in individuals with familial pancreatic cancer.
It would be very useful to group the risk factors according to the level of evidence:
- - very strong (such as cigarette smoking, chronic diabetes, obesity),
- - strong (such as poor oral health/oral hygiene, chronic pancreatitis, heavy alcohol drinking, dietary patterns with low intake of vegetables and fruits, no history of allergies, etc),
- - moderate (dietary patterns rich in meat and animal products, oral/gut microbiome, hepatitis B or C infection)
- - weak (such as environmental tobacco smoke, light to moderate alcohol drinking, physical inactivity, coffee, Helicobacer pylori infection)
Moreover, certain factors (such as tobacco smoking and diabetes) are excessively analyzed. Their review should be more concise.
The numbering of the headings and subheadings needs revision
Author Response
Dear Reviewer,
Thank you for the suggestions and time spent for performing this review. Please find response to the issues raised below. Changes in the manuscript are performed using: “track changes” tool.
Well-written paper on an interesting topic. However, several factors are missing, such as:
- - inherited genetic changes: mutations of BRCA1, BRCA2, PALB2, CDKN2A, ATM, MLH1, MSH2, MSH6, PMS2, STK11, and EPCAM. Also KRAS2, p16/CDKN2A, TP53, and SMAD4/DPC4. Genes associated with an increased risk of pancreatic cancer include BRCA1, BRCA2, P16, PRSS1, STK11/LKB1, hMLH1, hMSH2, FANC-C, and FANC-G.
- - Rare inherited conditions: Hereditary pancreatitis, Peutz-Jeghers syndrome, familial malignant melanoma and pancreatic cancer, hereditary breast and ovarian cancer syndrome, Lynch syndrome, Li-Fraumeni syndrome, familial adenomatous polyposis
The issues raised by the reviewer are of utmost importance. However in the submitted manuscript we wanted to focus on the risk factors of non-hereditary pancreatic cancer. We have addressed the complex clinical problem of hereditary pancreatic cancer and familial pancreatic cancer in our previous publication: “Current status of inherited pancreatic cancer” DOI10.1186/s13053-022-00224-2. That is the reason why we have omitted the topic in current review.
- - Cirrhosis
Thank you for that suggestion. In fact we have considered adding cirrhosis as an additional risk factor for pancreatic. However during our search we have noted that the risk may be attributed to the risk factors already described in the manuscript (alcohol consumption, tobacco smoking, microbiota – including H. pylori infection) and some of them are also related to increased risk of liver cirrhosis (predominantly ethanol). Therefore, we thought that are many confounders in this case and we focused “primary” causes/factors.
The authors should underline the importance of genetic testing in individuals with familial pancreatic cancer.
In the above mentioned paper regarding inherited pancreatic cancer we have included a summary of available recommendations on follow-up in patients prone to inherited/familial pancreatic cancer including: Peutz -Jeghers syndrome, familial atypical multiple mole melanoma, hereditary breast-ovarian cancer syndrome, Lynch syndrome, Ataxia-teleangiectasia syndrome, Hereditary pancreatitis, Familial pancreatic cancer
It would be very useful to group the risk factors according to the level of evidence:
- - very strong (such as cigarette smoking, chronic diabetes, obesity),
- - strong (such as poor oral health/oral hygiene, chronic pancreatitis, heavy alcohol drinking, dietary patterns with low intake of vegetables and fruits, no history of allergies, etc),
- - moderate (dietary patterns rich in meat and animal products, oral/gut microbiome, hepatitis B or C infection)
- - weak (such as environmental tobacco smoke, light to moderate alcohol drinking, physical inactivity, coffee, Helicobacer pylori infection)
According to Reviewer suggestions we have reshaped table 1. In order to provide a clearer stratification of risk factors we included specific data showing the increased risk of PC for a particular risk factor.
Moreover, certain factors (such as tobacco smoking and diabetes) are excessively analyzed. Their review should be more concise.
The numbering of the headings and subheadings needs revision
The numbering of subheadings has been corrected. The mistake occurred at the phase of transferring manuscript to Journal template.
Reviewer 2 Report
Authors described a review article about the risk factor for pancreatic cancer. They review 152 references and summarized appropriately.
However , Table 1 was a quotation from reference no. 127 which was review article of meta-analysis of 117 articles.
Authors were recommended to make and present a new table which they made from their review work.
Author Response
Dear Reviewer,
Thank you for the suggestions and time spent for performing this review. Please find response to the issues raised below. Changes in the manuscript are performed using: “track changes” tool.
Authors described a review article about the risk factor for pancreatic cancer. They review 152 references and summarized appropriately.
However , Table 1 was a quotation from reference no. 127 which was review article of meta-analysis of 117 articles.
Authors were recommended to make and present a new table which they made from their review work.
According to Reviewer suggestions we have reshaped table. In order to provide a clearer stratification of risk factors including those described in submitted paper. Currently it is presented in the form shown below:
Table 1. Summary of the PC risk factors including the magnitude of risk (provided in brackets)
|
Category |
Modifiable risk factor of PC (risk level) |
Non-modifiable risk factor of PC (risk level) |
|
Increased risk |
Chronic pancreatitis (3.0-16.0x) |
Age (50->70 years old) (6.8x) |
|
Tobacco smoking (up to 2.5x) |
Male sex (1.0-1.4x) |
|
|
High ethanol intake (up to 2.0x) |
Ethnicity (Afro-Americans vs. Caucasian) (1.3x) |
|
|
Acute pancreatitis (2.0x) |
Non “0” blood type (1.3-1.7x) |
|
|
Type 2 diabetes mellitus (1.5-2.0x) |
|
|
|
Periodontal disease (1.5-1.7x) |
||
|
Metabolic syndrome (up to 1.6x) |
||
|
HBV/HCV infection (1.4-1.5x) |
||
|
H. pylori infection (1.4-1.5x) |
||
|
Frequent red meat consumption (1.2x) |
||
|
Fructose (1.2x) |
||
|
Obesity - per 5kg/m2 (1.1x) |
||
|
Microbiome (pending) |
||
|
Occupational hazard (varied) |
||
|
Reduced risk |
Tea consumption (0.5x) |
|
|
Diet rich in fruits (0.6x) |
||
|
Allergy – nasal (0.6x) |
||
|
Physical activity (pending) |
Reviewer 3 Report
In this manuscript, the authors performed a comprehensive review of past studies on the risk factors of pancreatic cancer. This can be a valuable resource of the community to better understand this cancer. I suggest a minor revision considering the frequent writing and formatting issues.
Specific comments:
1. Abstract, line 21-22 "A cluster consisting .... For therapeutic lifestyle", please re-word the sentence.
2. Line 32-34 "Treatment of PC .... After established diagnosis", please re-word the sentence.
3. Line 81, what does significant 6% and nonsignificant 6% mean? Please re-word.
4. Line 172. The nearly 20 studies related to smoking are not "scarce"
5. The itemization across the manuscript seems odd. I repeatedly see '1.2' even when the items are completely different. Please make sure everything is correctly itemized across the manuscript.
6. Please carefully double-check the manuscript for typo and other writing issues.
Author Response
Dear Reviewer,
Thank you for the suggestions and time spent for performing this review. Please find response to the issues raised below. Changes in the manuscript are performed using: “track changes” tool.
In this manuscript, the authors performed a comprehensive review of past studies on the risk factors of pancreatic cancer. This can be a valuable resource of the community to better understand this cancer. I suggest a minor revision considering the frequent writing and formatting issues.
Specific comments:
1. Abstract, line 21-22 "A cluster consisting .... For therapeutic lifestyle", please re-word the sentence.
The sentence has been rephrased into:
“Special attention should be given to individuals with a vicious cluster consisting of metabolic syndrome, tobacco smoking and alcohol consumption”
2. Line 32-34 "Treatment of PC .... After established diagnosis", please re-word the sentence.
The sentence has been rephrased into:
“Therapeutic outcomes of PC remain unsatisfactory. Epidemiological data, even from high-income countries, show that only 11.5 % of patients survive 5 years after established diagnosis.”
3. Line 81, what does significant 6% and nonsignificant 6% mean? Please re-word.
The sentence has been rephrased into:
“Trend analysis showed a statistically significant 6% increase in PC risk for every 10 pack-years in men (aHR, 1.06; 95% CI, 1.03–1.10) but not in women (aHR, 1.06; 95% CI, 0.92–1.22).”
4. Line 172. The nearly 20 studies related to smoking are not "scarce"
The adjective “scarce” has been removed. And currently the sentence is as follows:
“In conclusion, the majority of the prospective studies and meta-analyses shows positive association between smoking and PC risk.”
5. The itemization across the manuscript seems odd. I repeatedly see '1.2' even when the items are completely different. Please make sure everything is correctly itemized across the manuscript.
The numbering of subheadings has been corrected. The mistake occurred at the phase of transferring manuscript to Journal template.
Please carefully double-check the manuscript for typo and other writing issues.
The manuscript has been re-checked in order to eliminate typos and improve grammar issues.
Reviewer 4 Report
The paper is well written, and the topic would interest the journal readers.
Only a few comments:
In Figure 1 replace “peutz-Jegers“ with Peutz-Jegers.
A native English speaker should revise the manuscript for fluency and correct a few errors.
Re-number correctly the sub-chapters.
Maybe the paragraph dedicated to HBV and HCV infection warrants a larger space.
Author Response
Dear Reviewer,
Thank you for the suggestions and time spent for performing this review. Please find response to the issues raised below. Changes in the manuscript are performed using: “track changes” tool.
The paper is well written, and the topic would interest the journal readers.
Only a few comments:
In Figure 1 replace “peutz-Jegers“ with Peutz-Jegers.
A native English speaker should revise the manuscript for fluency and correct a few errors.
The manuscript has been re-checked in order to eliminate typos and improve grammar issues.
Re-number correctly the sub-chapters.
The numbering of subheadings has been corrected. The mistake occurred at the phase of transferring manuscript to Journal template.
Maybe the paragraph dedicated to HBV and HCV infection warrants a larger space.
The topic of relationship between hepatotropic viral infections and pancreatic cancers is interesting and data regarding the subject are still emerging. We have added a section regarding potential pathophysiological mechanism responsible for such an increased risk of PC. However, available data are not definite. Therefore, we have not expanded the section in order to reduce to minimum speculative conclusions. Currently the section is as follows:
“Recent meta-analyses [100, 101] suggest a higher risk of PC in HBV-infected (1.39 times) and HCV-infected (1.5 times) individuals. PC risk is somewhat modified after taking into account additional epidemiological data such as diabetes, chronic pancreatitis, alcohol consumption and smoking in the case of HCV-infected individuals, or the region in which the study population resided (Europe vs Oceania) in the case of HBV-infected individuals. Precise mechanisms responsible for increased risk of PC in viral hepatitis are not fully explained. There are studies showing the presence of HBV core antigen in acinar cells [102] confirming viral invasion of pancreatic cells, which might similarly to hepatocytes be responsible for chromosomal instability [103]. Another factor that may contribute to elevated PC risk is connected with more common occurrence of chronic pancreatitis in patients with hepatotropic viral infection [104]. Nevertheless, further studies are necessary to substantiate potential causal link between those viral infections and PC.”